

# Opinion: The importance of historical and paleoclimate aerosol radiative effects

Natalie Mahowald[1], Longlei Li[1], Samuel Albani[2], Douglas Hamilton[3], Jasper Kok[4]

[1] Department of Earth and Atmospheric Sciences, Cornell University, Ithaca, NY, USA

[2] Department of Environmental and Earth Sciences, University of Milano-Bicocca, Milan, Italy

[3] Department of Marine, Earth and Atmospheric Sciences, North Carolina State, Raleigh, NC, USA.

[4] Department of Atmospheric Sciences, University of California, Los Angeles, CA, USA

*Correspondence to*: Natalie Mahowald (mahowald@cornell.edu).

**Abstract.** Estimating the past aerosol radiative effects and their uncertainties is an important topic in climate science. Aerosol radiative effects propagate into large uncertainties in estimates of how present and future climate evolves with changing greenhouse gas emissions. A deeper understanding of how aerosols interacted with the atmospheric energy budget under past climates is hindered in part by a lack of relevant paleo observations and in part because less attention has been paid to the problem. Because of the lack of information we do not seek here to show the change in the radiative forcing due to aerosol

changes, but rather just estimate the uncertainties in those changes. Here we argue that current uncertainties from emission uncertainties (90% confidence interval range spanning 2.8 W/m2) are just as large as model spread uncertainties (2.8 W/m2) in calculating preindustrial to current day aerosol radiative effects. There are no estimates for radiative forcing for important aerosols such as wildfire and dust aerosols in most paleoclimate time periods. However, qualitative analysis of paleoclimate proxies suggests that changes in aerosols in different past times are similar in magnitude to changes in aerosols between

preindustrial and current day, plus there is the added uncertainty from the variability in aerosols and fires in the preindustrial. From the limited literature we estimate a paleoclimate aerosol uncertainty for the last glacial maximum relative to preindustrial of 4.8 W/m2. The uncertainty in the aerosol feedback in the natural Earth system over the paleoclimate (last glacial maximum to preindustrial) is estimated to be about 3.2 W/m2/°K as a first estimate of the 90% confidence interval range. In order to assess the uncertainty in historical aerosol radiative effects, we propose a new model intercomparison project, which would

include multiple plausible emission scenarios tested across a range of state-of-the art climate models over the historical period. These emission scenarios would then be compared to the available aerosol observations to constrain which are most probable. In addition, future efforts should work to characterize and constrain paleoaerosol forcings and uncertainties. Careful propagation of aerosol uncertainties in the literature is required to ensure consideration of all the uncertainties.



## 1    Introduction

While CO₂ radiative forcing has been the most important driver of the observed climate warming, aerosol interactions with radiation and cloud properties represent the largest source of uncertainty in the Intergovernmental Panel on Climate Change (IPCC) estimates of present day to preindustrial radiative forcing (e.g. IPCC, 2021). This large uncertainty is driven by the heterogeneity of aerosols in space, time, composition, size and shape in the current climate, complexity in aerosol impacts on radiation and clouds, poor knowledge of aerosols in historical and past climate conditions, and how aerosol processes have changed over time (Albani et al., 2018; Bellouin et al., 2020; IPCC, 2021; Carslaw et al., 2017). Substantial research emphasis has been placed on understanding the direct changes in emissions from human activities (e.g. fossil fuel combustion); these emissions are estimated based on many processes, including emission factors for different industries, estimates of the changes in location and intensity of different industrial facilities, as well as time dependent changes in technologies (e.g. (Bond et al., 2007; Klimont et al., 2017).  On the other hand, the radiative forcing due to changes in aerosol sources that are usually considered natural, such as from dust storms or wildfires, are arguably less well understood (Carslaw et al., 2010; Mahowald et al., 2011a; Regayre et al., 2018).  Because of limited paleo-observations, we currently rely on emission models that were calibrated using current observations and apply them to past climates, not only for industrial sources but also for wildfire and dust aerosols (Van Marle et al., 2017; Turnock et al., 2020; Gidden et al., 2019; Zhao et al., 2022).

However, the dominant mechanisms for natural emission processes are unlikely to have remained constant over time, and thus using present day observations to constrain past model predictions is biased towards the present day and thus anthropogenic influenced aerosol behavior (e.g. Hamilton et al., 2018). We therefore advocate that past model simulations should, as much as possible, be constrained using paleoenvironmental archives of past aerosol. Note that for IPCC estimates, most models use the same emission dataset(s) to drive emissions meaning that the aerosol radiative forcing uncertainty estimates based on Climate Model Intercomparison Project (CMIP6) estimates tend to accurately represent the uncertainty due to model spread using the same emission dataset, but do not necessarily aim to include the uncertainty due to emission uncertainties (Gidden et al., 2019; Bellouin et al., 2020; Thornhill et al., 2020; Pincus et al., 2016), which are especially large in preindustrial or paleoclimate climates (Li et al., 2019; Kok et al., 2023).

Here we argue that an improved characterization of the evolution of 'natural' aerosols, especially dust and wildfire aerosols, is needed to improve our understanding of aerosol radiative effects over the historical and paleoclimate



time periods. Before we can seek to constrain the uncertainties (e.g. Bellouin et al., 2020), we need to first characterize the different sources of uncertainties and their magnitude(Carslaw et al., 2017), especially emissions of natural aerosols (e.g. Hamilton et al., 2018; Kok et al., 2023), and propagate these uncertainties into the climate simulations. Additionally, we argue that radiative perturbations due to changes in natural aerosols that are affected by human actions, such as dust and wildfire aerosols, should be treated as a radiative forcing. Note that we use the IPCC glossary definition of radiative forcing, as the change in the top of atmosphere radiative balance due to the addition of a species (IPCC, 2021), and use the term radiative forcing instead of effective radiative forcing, for which the same arguments apply. In our discussion here, we include both direct effects and aerosol-cloud interactions, realizing that the largest uncertainties are often associated with aerosol-cloud interactions. We discuss 1) the limited available data constraining the changes in dust and wildfire aerosol emissions over the historical time period and how those compare to the CMIP6 emission datasets and whether natural aerosols can be considered climate feedbacks; 2) consider whether aerosols can be considered feedbacks or forcings; 3) propagating the uncertainty in emissions of natural aerosols to create more realistic estimates of aerosol radiative effect uncertainties, 4) estimates for the uncertainties in aerosol radiative effects for paleoclimate time periods, 5) other sources of aerosol uncertainties and 6) the importance of accurate aerosol radiative effect uncertainties for climate science. Finally, 7) we propose a path forward to improve the characterization of the uncertainties in aerosol radiative effects, which would then allow us to start constraining the uncertainties with observations.

## 2 Observations of natural aerosol changes since 1850

Natural aerosols include dust, wildfire emissions, sea salts and biogenic organic aerosols, among others. Aerosols such as dust or carbonaceous and sulfur species emitted by wildfires can be generated under natural conditions. As such, these aerosols can produce important feedbacks in a changing climate (Allen et al., 2016; Kok et al., 2018; Thornhill et al., 2021). However, some of these aerosols, notably dust and wildfire aerosols, are also directly or indirectly affected by human actions. For instance, dust emissions can be increased substantially both by human disturbance of the natural landscape and by anthropogenic diversions of surface water flows (Lee et al., 2012; Ginoux et al., 2012; Xi and Sokolik, 2016). Similarly, wildfire emissions can be increased by open fires set by humans as part of deforestation or agricultural practices (van der Werf et al., 2010). Additionally, natural wildfire emissions can be suppressed by human activity, for example active firefighting or removal of forests and grasslands due to agricultural and urban development (Knorr et al., 2014; Bistinas et al., 2013). As such, the radiative perturbation due to historical changes in these natural aerosols can be partially due to both human land use changes (a forcing) and natural and anthropogenic climate changes (a feedback).



## 2.1 Historical changes in desert and agricultural dust

While the global concentration of long lived and well-mixed species such as $CO_2$ can be retrieved from ice cores, aerosols in the troposphere have lifetimes of only a few days to two weeks (Textor and others, 2006), meaning that aerosol observations need to be available across the globe in dozens or hundreds of locations at a high temporal frequency before we can begin to build a reasonable understanding of their distribution and behavior. Unfortunately, for most aerosol species we do not have that kind of coverage for most time periods, including the

present day (e.g. see spatial coverage in Naik et al., 2021). Dust is an exception in some respects, in that dust is preserved to some extent in several natural archives, and thus there are compilations of dust changes over different time periods. For paleoclimate conditions (e.g. the last glacial cycle, the Holocene, etc.) the main natural archives for dust include ice cores and marine and terrestrial (loess/paleosol) sediment records, for which paleodust compilations exist (e.g. Albani et al., 2015). Because of their nature, most of those natural archives do not have the

potential to cover the last few centuries including the late Holocene to preindustrial to modern transitions, due to issues with dating or disturbance of surface sediments (e.g. for many marine sediment records the core top is lost during retrieval and thus the last 100 or so years are not easily obtainable; loess provides the substrate for very fertile soils exploited for agriculture in the last millennia) (Maher et al., 2010). Other archives, such as in particular firn cores from the polar areas, ice cores from mountain glaciers, and ombrotrophic peat bogs, have the potential

to preserve dust deposition records over the last decades/centuries, although they are still affected by major uncertainties when it comes to retrieving accurate deposition records (Albani et al., 2015). With these caveats in mind, there is still paleo data that can be compiled to infer the evolution of desert dust in different regions across the globe since the pre-industrial (Kok et al., 2023).

Desert dust is generated in dry, unvegetated regions with strong winds, and the generation of dust may be enhanced in regions with human land use (Ginoux et al., 2012). Between the 1960s and 1980s dust was observed to have changed by a factor of 4 over the North Atlantic region, perhaps due to expansion of land use, due to precipitation changes during the Sahel drought, or due to changes in winds (Mahowald et al., 2002; Prospero and Lamb, 2003; Evan et al., 2016). Paleoclimate evidence also suggests that dust is very sensitive to both climate change as well

as land use (Lambert et al., 2008; Neff et al., 2008; Mulitza et al., 2010).

A recent synthesis of dust deposition observations suggests a $55 \pm 30\%$ increase in dust globally since preindustrial times (Kok et al., 2023). While there is not sufficient data for each source to have complete confidence in such estimates, they still represent our best knowledge of the state of changes in dust since 1850s. Unfortunately, the





model simulations from the last Climate Model Intercomparison Project (CMIP6) do not match these changes, either because of the lack of correct inclusion of the impact of land use onto dust, or incorrect simulation of the feedbacks of natural and anthropogenic climate changes onto dust (Fig. 1).  The CMIP6 models show relatively constant dust amounts in contrast to the large change that is observed (Kok et al., 2023).

## 2.2 Historical changes in wildfire and open fires


Paleoenvironmental archives for wildfires and other open fires include charcoal records near the source regions (Marlon et al., 2008), ice cores (McConnell et al., 2007), tree ring scars, and speleothems.  While wildfire is a natural ecosystem processes , humans can modify the frequency and intensity of fires through many mechanisms (Bowman et al., 2009, Pechony and Shindell 2010). Humans are responsible for much of the ignition of fires today,

but also convert land from natural to managed land, reducing the area available for wildfires   In addition, humans may suppress fires once they start (Bowman et al., 2009; Kloster et al., 2010).  Observations over the last 20 years from satellites show strong interannual variability with 50% changes in emissions observed correlated with climate signals, although human contributions could have played a vital role(van der Werf et al., 2006) .  It has long been the assumption in the atmospheric chemistry and aerosol community that open fires have increased due to humans

(Knorr et al., 2014; Bistinas et al., 2013).   The open fire emission data used for the CMIP6 uses data from current satellites to predict emissions from 1997 onwards, however, prior to the Satellite fire Era the emissions are generated from fire models with limited additional proxy evidence. These models include a range of representations of how fires have evolved with increasing human population density, and not all account for active or passive fire suppression. Overall, this results in CMIP6 fire emissions   increasing since 1850s (Van Marle et al., 2017).

However, there is ample evidence that the relationship may be much more nuanced.  Paleoclimate data from charcoal records suggests a maximum in open fires in the 1850s, and a decrease since then (Marlon et al., 2008). Satellite data shows a global decrease in burned area over the last decades, driven primarily from the conversion of natural lands to agricultural and pastoral lands  (Andela et al., 2017, Jones etal. 2022).  Indeed, those fire models which include a more realistic representation of how fires and human population density are related simulate a

much higher amount of fires during 1850 than the CMIP6 (or AEROCOM) emission dataset suggests, and including these fire emission simulations in aerosol models improves the match to the available ice core data of the deposition ratio between present day to preindustrial of black carbon (yellow and red symbols) than using the default CMIP6 datasets (blue symbols) (Hamilton et al., 2018; Liu et al., 2021) (Fig. 2).  This suggests that is probable that there were more open fire emissions during the PI than accounted for in the emission inventories used for CMIP6

simulations (Gidden et al., 2019).



Is it plausible that emissions from wildfires and other open fires, such as agricultural fires, could be so much larger in preindustrial times than currently being accounted for in climate model emission datasets? It is difficult to know for certain of course, but a few examples show the possibility, using different mechanisms. One study suggests that higher fire amounts during 1850 could be due to less land use change, reducing the natural area available for fires (van der Werf et al., 2013) which today is causing a decrease in fire burn area (Andela et al., 2017). Other studies have suggested that wildire suppression has been important for reducing fires in North America for example over the last 50 years (Marlon et al., 2012) . In addition, agricultural open burning could be important; today, northern India represents the region with some of the highest aerosol optical depths and worst air quality (Li et al., 2022; Burnett et al., 2018).  Despite the area having large population centers and industrial emissions, one of the largest sources in that region is agricultural burning (Cusworth et al., 2018). Similarly, the ban on straw burning, a primary source of pollutants in central and eastern China (Wu et al., 2018), has become a national policy for air pollution control, as in many other countries. This suggests that indeed, the high emission factors of open burning make it a very effective source of aerosols.

In addition, wildfires and open fires represent some of the most important aerosols for direct and aerosol-cloud radiative effects, with a total radiative effect in the current climate of -2 W/m$^2$ (IPCC, 2021; Penner et al., 1992). Changes in wildfires and open fires represent about -1 W/m$^2$ or  50% of the anthropogenic aerosol radiative forcing since 1850 (Unger et al., 2010; IPCC, 2019, 2021).The overall negative forcing results from the effect of an increasing aerosol burden increasing cloud albedo, and from the prevailing effects of preferentially scattering sulfates and particulate organic matter (secondary aerosol from fire emission of precursors such as biogenic volatile organic compounds), over the preferentially absorbing black carbon emissions from fires (Hamilton et al., 2018; Carslaw et al., 2017; Penner et al., 1992; IPCC, 2021).

In summary, tradeoffs between the effects of climate change and land use make it difficult to estimate past changes in the loading of dust, and smoke from open fires. Changes in other aerosols (fossil fuels, biofuels, biogenic aerosols, sea spray), are difficult to estimate as well. Without observations of these aerosols in past climates, it is difficult to have confidence in our existing emission models and their past and future emissions projections.

## 3   Are dust and other natural aerosols forcers or feedbacks?



The nomenclature that dust and wildfire aerosols are natural aerosols is perhaps misleading and might have caused the important changes that have occurred in these aerosols to receive insufficient consideration by climate scientists. For instance, in the Sixth Assessment Report, the radiative perturbation due to the ~50% increase in dust over the historical record (Hooper and Marx, 2018; Mahowald et al., 2010; Kok et al., 2023) is not explicitly accounted for as a radiative forcing of the climate system, although the dust-climate feedback was quantified (Forster et al., 2021; Naik et al., 2021). There are several reasons why not explicitly accounting for the historical dust increase as a radiative forcing could be problematic. First, although the exact proportion of modern dust that can be considered anthropogenic is uncertain (Tegen et al., 2004; Mahowald, 2007; Ginoux et al., 2012; Stanelle et al., 2014), a large body of work indicates that human land use changes in semi-arid and arid lands can produce a large increase in dust aerosol emissions (Neff et al., 2008; Webb and Pierre, 2018). Such land use changes have been widespread since the Industrial Revolution (Klein-Goldewijk, 2001) making it likely that a substantial part of the historical dust increase – perhaps even most of it – was driven by human land use changes (Ginoux et al., 2012; Hooper and Marx, 2018; Kok et al., 2023), which thus constitutes a radiative forcing.

A second reason why not explicitly accounting for the historical dust increase as a radiative forcing might be problematic is that this implicitly assumes that the historical dust increase has been due to a climate feedback. However, the dust change per degree global surface temperature warming is inconsistent between time periods. Indeed, the dust increase during the planetary warming of the past century is opposite to what is seen in the paleo-record, for which cold periods like the Last Glacial Maximum coincide with high dust loadings (Albani et al., 2014). Moreover, there is no model consensus on whether dust will increase or decrease under future climate warming, in part because of large uncertainties in how precipitation in arid regions will change (IPCC, 2019). This inconsistency in the dust change per unit global surface temperature warming could be due to a number of factors: (i) the historical dust increase was primarily driven by human land use changes, not climate changes; (ii) the dust feedback is highly dependent on either the climate state or the timescale; or (iii) the dust feedback occurs over much longer timescales than the observed ~century-scale dust increase. Whatever the reason, the fact that the dust change per unit surface temperature change is not consistent between time periods undermines both the plausibility and the usefulness of classifying historical dust changes as a feedback in the context of future climate predictions.

Finally, and most importantly, the historical increase in dust that is indicated by dust deposition records (McConnell et al., 2007a; Mulitza et al., 2010; Mahowald et al., 2010; Hooper and Marx, 2018) is not captured by climate models (Kok et al., 2023) (Fig. 1). These models therefore also predict a dust-climate feedback that is





indistinguishable from zero (Thornhill et al., 2021; Kok et al., 2023). As such, not explicitly accounting for dust changes as a radiative forcing has the net effect of omitting this potentially important perturbation to Earth's energy
balance. This can bias climate sensitivity constraints and projections of future climate changes (Kok et al. 2023). Because it is unlikely that the anthropogenic component and the climate feedback component of historical dust loading change can be reliably separated in the near future, the best remedy to this problem would be for future climate assessments to explicitly treat the historical dust change as a radiative forcing. Furthermore, because climate models currently cannot reproduce the historical dust changes (Kok et al., 2023; Mahowald et al., 2010),
we also recommend that climate models use a dust reconstruction to externally force the historical dust increase.

If aerosols are considered feedbacks, the full uncertainty in the feedback should be included. For example, from this analysis, the feedback uncertainty should be +/- 1.6 W/m$^2$ uncertainty from preindustrial to current (see Section 4), over which time temperatures increased by about 1/°K, so that means an uncertainty in the feedback of +/- 1.6
W/m$^2$/°K (90% confidence).

## 4        Characterizing preindustrial to current aerosol forcing uncertainty

Since CMIP6 aerosol simulations are not consistent with available observations for dust and open fires, it is clear
that additional uncertainty needs to be added to the aerosol radiative forcing estimates for the preindustrial to present day in order to make sure that the uncertainty range includes available observations. This is a substantial undertaking, but here we show schematically a back-of-the-envelope calculation of how including the observations would affect estimates of aerosol radiative forcing uncertainty (Fig. 3).

We use here a slightly different nomenclature than (Sherwood et al., 2020) for example, to emphasize the uncertainties in radiative forcing, without introducing too much nomenclature, and thus define ΔF as the change in radiative forcing between two different times, and Σ is the uncertainty in that estimate using the 90% confidence intervals.

If we sum the sources of uncertainty currently available in the literature ($\Sigma_{Fires}^{PD-PI} = 2.8$ W/m$^2$ from fires, $\Sigma_{Dust}^{PD-PI} = 0.4$ W/m$^2$ from dust and here we assume a10% error for industrial emissions for $\Sigma_{Industry}^{PD-PI} = 0.2$ W/m$^2$ (Hamilton et al., 2018; Wan et al., 2021; Kok et al., 2023) using Eq. (1), we obtain 2.8 W/m2, clearly dominated



by fires, which makes sense, since much of the anthropogenic radiative forcing is from this source (Unger et al., 2010) (see Table 1 for terms).


$$\Sigma_{Emis}^{PD-PI} = \left[ (\Sigma_{Fires}^{PD-PI})^2 + (\Sigma_{Dust}^{PD-PI})^2 + (\Sigma_{Industry}^{PD-PI})^2 \right]^{1/2} \ , \tag{1}$$

The uncertainty in radiative forcing from uncertainty in preindustrial emissions ($\Sigma_{Emis}^{PD-PI} = 2.8$ W/m$^2$) is similar in magnitude to the uncertainty from using one emission scenario for the historical time period (2.8 W/m$^2$ which is

the unconstrained model uncertainty using 90% confidence intervals) (Bellouin et al., 2020; Sherwood et al., 2020), which we refer to here as the unconstrained aerosol process uncertainty for present day to preindustrial ($\Sigma_{Process}^{PD-PI}$). The model spread in radiative forcing with the same emission scenario is due to differences in model simulations of concentration, radiation and cloud interactions using the same emission change, which are large, because these processes are poorly understood (Li et al., 2022; Pincus et al., 2016). The models make different assumptions about

aerosol lifetime, size distribution, aerosol microphysics, which results in different radiative forcings, so we assume this spread in models is the uncertainty in processes ($\Sigma_{Process}^{PD-PI}$), which is 2.8 W/m2. One could also think of this uncertainty in the process as coming from two sources: variability in present day processes in simulating aerosols (e.g. that radiative forcing is sensitive not just to the total emissions, but also to where, what kind and what else is in the region (Li et al., 2022; Bellouin et al., 2020) and one part that is proportional to the strength of the change in

aerosols, which could be proportional to the change in radiative forcing (ΔF) times some factor $\gamma$ (Eq. 2).

$$\Sigma_{Process}^{PD-PI} = \left[ (\Sigma_{Process}^{PD})^2 + (\Delta F * \gamma)^2 \right]^{1/2} \ , \tag{2}$$

We propose that future studies should identify the strength of the base uncertainty ($\Sigma_{Process}^{PD}$) and the portion of this

process uncertainty that is proportional to the strength of the change in the radiative forcing ($(\Delta F * \gamma)^2$).

The total unconstrained uncertainty due to aerosol changes could be estimated as being 4 W/m$^2$ using equation 3, assuming the uncertainties are orthogonal (Eq. 3).

$$\Sigma_{TotalUNC}^{PD-PI} = \left[ (\Sigma_{Process}^{PD-PI})^2 + (\Sigma_{Emis}^{PD-PI})^2 \right]^{1/2} \ , \tag{3}$$



Emissions from industry (which are likely better known) have been increasing since 1850 in the CMIP6 simulations

from which we estimated the aerosol radiative forcings. But aerosol from wildfires have also been increasing during this historical period in these simulations (Gidden et al., 2019). As discussed in Section 1, the paleoclimate data (and fire models which explicitly account for passive fire suppression effects of land use change) suggest that open fires have been decreasing since 1850, potentially offsetting the increase in industrial emissions (Fig. 4b). In contrast, the CMIP6 wildfire emissions assume large increases since 1850 in wildfires and open fires (Van Marle

et al., 2017). This produces a large uncertainty in the time series of aerosol forcing over the historical period. Notice that since wildfire emissions can vary strongly over a couple of years or decades (van der Werf et al., 2004), it should not be assumed (without observations) that the radiative forcing from wildfires follows either the top or the bottom of the error bar, but rather could vary from one year to the other over the whole range.

Some of the difficulty of looking at preindustrial to present day aerosol changes is simply understanding what 'natural' aerosols would look like without humans. Unfortunately, there are strong fluctuations across the time period just before the industrial era (e.g. 1500-1850) in fires (Fig. 5), some potentially associated with humans (e.g. perhaps the increase in 1850 and decrease after this time period), but a large change during the little ice age suggests that climate change can radically change the fires (van der Werf et al., 2013). While IPCC has used 1850 to 1900

as the preindustrial period for climate simulations (e.g. Allen et al., 2018), for aerosols this is not an ideal time period, as it is likely that aerosols are already elevated due to anthropogenic activities during this time period, while 1750 could be better, although still part of the little ice age. The issue of what is the right baseline for preindustrial aerosols is important also for considering paleoclimates (Section 5). As shown in Fig. 3 and 5, the uncertainty in preindustrial to present day emission changes in aerosols is driven by preindustrial emission uncertainties, which

is partly associated with variability across the preindustrial time period.

There are, of course, constraints on present day radiative effects from aerosols from satellites and other tools, which can constrain the last 30-40 years (e.g. Bellouin et al., 2020). And there are energy constraints on the present day to preindustrial change in aerosol radiative forcing using energy balance constraints (Smith et al., 2021; Sherwood

et al., 2020) which result in an 57% reduction in the uncertainty using fixed emissions ($\Sigma^{PD-PI}_{Const\_Process}$=1.6 W/m²). Unconstrained emission uncertainties and unconstrained process uncertainties have yet to be combined in a rigorous method like Bellouin et al. (2020) did for process uncertainties (in that study they assume that emissions are well





known), but this should be done in the future. Adding in the uncertainties in emissions, especially from wildfires, would mean that while directly emitted anthropogenic aerosols are going up (as estimated in CMIP6), wildfire

emissions may be going down. The wildfire aerosols resulting from these emissions would thus partially offset the radiative cooling from the increase in anthropogenic aerosols. Thus, if we take the case of high wildfires in the preindustrial (-2 W/m$^2$ in Fig. 4), this could imply that estimates of anthropogenic aerosol radiative forcing today which are large (-2W/m$^2$) would be more likely. This would have important implications for climate warming over the next few decades, as anthropogenic emissions of aerosols are likely to decrease, leading to more warming than

projected without including preindustrial aerosol emission uncertainties.

## 5        Characterizing paleoclimate aerosol forcing uncertainty

Unfortunately, except for dust or wildfires in certain time periods (Albani et al., 2015; Power et al., 2007; Zennaro

et al., 2014; Marlon et al., 2008), there is very little information about the distribution or amount of aerosols in different climate regimes, and therefore we do not know the emissions well, nor then the impact of those emissions onto climate.  We can envision these uncertainties are mostly unknown unknowns.  We have some information that they are likely to be large (since aerosol uncertainties today are so large, and we know less about paleoclimate aerosols), but we cannot yet directly constrain these.  We do know that there were large fluctuations: for example,

dust was likely 2-4x higher in the last glacial maximum than today (Lambert et al., 2015; Mahowald et al., 1999; Albani et al., 2014; Albani and Mahowald, 2019), while between preindustrial and present day, the change is smaller at only approximately 2x (Kok et al., 2023) (Fig. 6).  For dust, we have estimates at the last glacial maximum (LGM) and 6000 bpa (Albani and Mahowald, 2019; Albani et al., 2014), which suggest that the changes in radiative forcing could be on the order of 0 to -2 W/m$^2$ (Albani et al., 2018), although studies using carefully

compared dust optics show smaller radiative forcings, because dust both absorbs and reflects both short and long wave radiation  (Albani and Mahowald, 2019; Braconnot et al., 2021).

But changes between preindustrial and present day aerosol radiative forcings are dominated by changes in fires (Section 1 and 2): are these changes are large in the paleodata as seen in the last 150 years?  The limited paleodata

suggests large changes in fires during different time periods in the past (Fischer et al., 2015; van der Werf et al., 2013; Zennaro et al., 2014; Arienzo et al., 2017). For example, in considering cold periods like the last glacial maximum, there is likely a large reduction in fires in high latitudes, due to the presence of the Laurentide and Fenno-Scandinavian icesheets, which is consistent with fire proxies in Greenland ice cores (e.g. the ammonium



record for the North Greenland Ice core Project (Fischer et al., 2015) . Generally, the charcoal record suggests

lower fire frequency in the last glacial maximum than preindustrial (Marlon et al., 2016), although ice sheets could

have removed sediment records of wildfires (Fig. 6).  For climate impacts, the low and mid latitude fires tend to be

more important today (Hamilton et al., 2018), so more information on the frequency and extent of wildfires in those

regions are the most important, and difficult to retrieve from ice cores. The changes seen in wildfires between

preindustrial and for example, last glacial maximum are as large if not larger than those seen between preindustrial

and present day (Fig. 6). These studies suggest qualitatively that the changes we have seen in fires over the

preindustrial to present day are not unprecedented in size, but rather are similar to paleoclimate changes.

In addition, paleoclimate data such as temperature changes or aerosol changes are done relative to preindustrial

changes, and as discussed in Section 2, and shown in Fig. 5, there is substantial variability in preindustrial fires.  It

is unclear what value to use for preindustrial aerosol emissions to compare to paleodata values: do we use 1850

values or values from the little ice age? Or some average?  One can think of the variability in changes in emissions

between some time (T) and PI as being shown in Equation 4. The uncertainties in emissions in PI are driving the

uncertainties in PD-PI describe above and are about 2.8W/m$^2$ ($\Sigma_{Emis}^{PI}$): increased knowledge is unlikely to reduce

these uncertainties from variability.  In addition, the changes in emissions between PI and any other time period

are likely to be similarly large, but may not be orthogonal ($\Sigma_{Emis}^{T}$).  Adding these together (using Eq. 4), we obtain

$\Sigma_{Emis}^{T-PI}$=4.0 W/m$^2$.  Note that the uncertainties in emissions proposed here for different time periods could be

constrained to some extent $\left(\Sigma_{Emis}^{T}\right)$, but uncertainties due to the variability in PI emissions (Fig. 5) would be

difficult to constrain, and there may be substantial variability as well as uncertainty in emissions during different

time periods, so the values proposed here may actually underestimate the uncertainty.


$$\Sigma_{Emis}^{T-PI} = \left[(\Sigma_{Emis}^{PI})^2 + (\Sigma_{Emis}^{T})^2\right]^{1/2} , \tag{4}$$

Once we have paleoproxies to provide data about changes in fire emissions, especially, we can constrain the

emission uncertainties for paleo time periods relative to present day, hopefully. Unfortunately knowing the

emissions does not translate into knowing the radiative forcing in past times, as we known from our experience

simulating preindustrial to present day emission changes in existing models using the same emissions (Bellouin et

al., 2020). There are uncertainties of translating these changes in emissions into changes in direct radiative and




aerosol-cloud interactions or process uncertainties, which we assume here, since we do not have better information,
that these are a similar size to present day to preindustrial uncertainties ($\Sigma_{Process}^{PD-PI}=\Sigma_{Process}^{T-PI}$= 2.8 W/m²). These
uncertainties are due to differences in the modeling of aerosols, and assumptions about size and how aerosols
interact with clouds which can be different depending on where the aerosols are emitting: this uncertainty will
remain in paleoclimates, and might even become larger, since the aerosol size, composition and mixing state could
be quite different and the very important impact of aerosols onto clouds is sensitive to the background conditions
(Carslaw et al., 2017). Fires from different ecosystems, or even different types of fires in the same ecosystems,
have very different emissions of black carbon, organic carbon and sulfate, and thus different effects, but we do not
know how these will change in different time periods. Natural aerosols are the source of much the uncertainty in
today's climate compared to anthropogenic aerosols, because of the difficulty of the estimating the exact timing
and distribution of emissions, as well as the sources are more complicated in composition and location (e.g.
Mahowald et al., 2011b; Carslaw et al., 2017; Rathod et al., 2020). Similar to the present day relative to
preindustrial, we can estimate the paleotime to preindustrial radiative forcing uncertainty using equation 3 and
obtain 4.8 W/m² as the range of uncertainty for paleotime periods. (Notice that if we can constrain the change in
radiative forcing from changing emissions of aerosols to be smaller than that between preindustrial and present
day, using obsevations and equation 2, it is possible we could proportionately reduce the uncertainty in radiative
forcing from aerosols from process uncertainties, see equation 2.) Converting this radiative forcing uncertainty into
a feedback uncertainty requires knowing the temperature change, which is also uncertain, but if we use 3C as a
reasonable value, the aerosol feedback uncertainty derived from the last glacial maximum to PI is +/-1.6W/m²/°K,
similar to the value derived from the PD-PI time period. If we add in the uncertainty in temperature change between
preindustrial and last glacial maximum, this estimate would be even larger, of course.

It seems likely that aerosol emissions from fires during the last glacial maximum are much smaller than
preindustrial or present day values, while estimates suggest dust is ~3x larger in the last glacial maximum than the
present day. Will these changes in aerosols balance out? That is unlikely but vital to consider. Dust is by mass the
most important aerosol in the atmosphere, and contributes substantial to direct forcing and ice nucleation processes,
but fire emissions are important for liquid aerosol-cloud processes (Mahowald et al., 2011a; Carslaw et al., 2010).
Because of the non-linearity in aerosol-cloud interactions, small changes in fire emissions in pristine environments,
like the last glacial maximum, might be even more important than estimated here (Carslaw et al., 2017).
Understanding the aerosol interactions with clouds especially for the last glacial maximum is both important and
intriguing.




Since today the largest uncertainties in the radiative forcing come from aerosol uncertainties, estimates in past climates should ensure that aerosol uncertainties remain one of the largest aerosols in those times as well: how could we know the change of aerosols from some paleoclimate to preindustrial better than we know the change in aerosol forcing between preindustrial to present day? If ice sheets or insolation or continental distributions are

different and causing large changes in top of atmosphere fluxes and thus climate regime, most likely aerosol changes are equally large, but we do not know in what direction. More analysis might result in even larger changes in radiative forcing and its uncertainty in some time periods, since here we are assuming, without prior information, that the radiative forcing of any paleotime period relative to preindustrial is around 0.0 W/m$^2$. For the LGM, for example, if we only include dust, a more negative value should be chosen as the mean, since we have evidence of

increased dust in the paleorecord (Albani et al., 2018). On the other hand, the limited data suggests that fires have substantially decreased relative to preindustrial, which would warm the climate. It is beyond the scope of this opinion piece to characterize the central estimate, but rather here we just point out the many uncertainties in these estimates.

**6 Aerosol processes and other sources of uncertainty**

So far here we have focused on the more frequently studied processes of aerosol direct radiative effects, and aerosol-cloud interactions with an emphasis on cloud condensation nuclei. However there remain substantial uncertainties in these aerosol radiative effects even in the current climate (Bellouin et al., 2020; Li et al., 2022) Aerosols are

spatially and temporally hetereogeneous in composition, size and amount, leading to vastly different physical and chemical properties. They are in general poorly observed compared to metereological phenomenon (e.g. Naik et al., 2021). Not only the bulk composition matters but the details of the mixing state and size are vital for radiative and cloud interactions (Matsui et al., 2018; Bond et al., 2013; Li et al., 2022, 2021). In addition, in preindustrial times, the impact of aerosols, for example on cloud properties, can be different than present day because of a lower

background aerosol amount (Carslaw et al., 2017; Hamilton et al., 2014) .

The impacts of large changes in important ice nuclei such as dust or primary biogenic particles is likely to be large but has yet to be fully assessed (Burrows et al., 2013; Murray et al., 2021; Storelvmo, 2017). Another important feedback that is relatively well know but not included in most climate models is due to nitrogen aerosols such as

ammonium or nitrate (e.g. Bauer et al., 2007; Paulot et al., 2016). Future concentrations of aerosols deriving from





land use practices such as ammonia or nitrate are not likely to decrease as quickly as from fossil fuels(Gidden et al., 2019). Indeed, as sulfate is phased out, more nitric acid will form nitrate aerosols (due to higher pH), partially buffering decreases in aerosol AOD (Paulot et al., 2016; Pye et al., 2009) . Including better parameterizations of ice nucleating particles and nitrogen aerosols is key to improving future aerosol projections.


In addition, aerosols can provide nutrients and pollutants to different ecosystems (Mahowald et al., 2017; Hamilton et al., 2021), linking aerosol changes to changes in biogeochemistry and the carbon cycle. These effects could potentially be quite large (0.5 W/m$^2$ +/0.4W/m$^2$) (Mahowald, 2011), but are poorly constrained, and do not explicitly appear in the standard radiative forcing diagram, since they reflect $CO_2$ that is not in the atmosphere, but

could have been (Mahowald, 2011).

Another natural emission to which estimates of radiative forcing are sensitive to is biogenic volatile organic compound emissions (BVOCs). BVOCs are a major source of new aerosol particles in the atmosphere. (Guenther et al., 2006; Arneth et al., 2010). Furthermore, biogenic particle formation processes contributed more to the aerosol

burden in the PI than the PD (Gordon et al., 2016). Estimates of the radiative forcing of BVOC are sensitive to how well characterized new particle formation processes are in a model. The recent addition of an organic particle formation pathway, which occurs in the absence of $SO_2$ (such the PI), results in an increased aerosol burden in the past than the present. Once more increasing the PI aerosol burden reduces the estimate of the aerosol forcing over the historical period, this time by reducing the cloud forcing by ~0.2 W/m$^2$ (Zhu et al., 2019).


## 7        Implications of including uncertainty in emissions in radiative forcing estimates

Aerosol radiative forcing and its uncertainty is used extensively in climate change science, including to constrain climate sensitivity (Sherwood et al., 2020) and thereby future climate changes (IPCC, 2021). Because the published

aerosol uncertainties tend not to include poorly constrained uncertainties such as discussed here, this information is not effectively passed to physical climate scientists who use these estimates. For example, a recent review of climate sensitivity (Sherwood et al., 2020) focused on using independent methods to reduce uncertainty in climate sensitivity. In that paper, aerosol radiative forcing uncertainties for different time periods are mentioned in several different places. They use the unconstrained model range of the aerosol radiative forcing obtained by (Bellouin et

al., 2020), which as discussed above, does not account for emission uncertainties. Paleoclimate constraints are often used for constraining climate sensitivity, as discussed in (Sherwood et al., 2020). Currently there exist



estimates for dust aerosol radiative forcing changes between last glacial maximum and current, which is included in (Sherwood et al., 2020) as -1.0 +/- 1.6 W/m$^2$ (90% confidence intervals: they report 1 sigma values of +/- 1 W/m$^2$ in Section 5.2.2 which are converted to 90% confidence here by multiplying by 1.6 as a first estimate) but

no mention is made of the potential for changes in the more important wildfires. As discussed in Section 4, estimates for radiative forcing of aerosols for paleotime periods especially wildfires are missing, but should be estimated to be 0.0 +/- 2.4 W/m$^2$ (90% confidence interval). In addition, the aerosol feedback within the system is assumed in Sherwood et al., 2020 to have an uncertainty of +/- 0.22 Wm$^2$ (they report 1 sigma values of +/-0.15 W/m$^2$ in section 3.2 which here we convert to 90% confidence intervals) whereas here we estimated the aerosol

feedback uncertainty to be +/-1.6 W/m$^2$ (90% confidence), substantially larger. Including more realistic aerosol uncertainties into estimates of climate sensitivity should be done to ensure adequate propagation of errors, although they are unlikely to change the central estimates (Sherwood et al., 2020).

In addition, some authors argue that there were not significant changes in aerosol radiative forcing during the 1970s

and 1980s, using standard CMIP6-type estimates, and try to estimate climate sensitivity in this time period (Jiménez-de-la-Cuesta and Mauritsen, 2019). As noted above, however, the 1970s is a time period of the Sahel drought, and dust radiative forcing between 1960s and 1980s changed by perhaps -0.57 +/- 0.46 W/m$^2$ (Mahowald et al., 2010), suggesting that is not an ideal time period to target. One should add onto this estimate the possibly important changes in wildfires which could have occurred over this time period but for which we do not have data.


In addition to climate sensitivity, some studies use the CMIP6 simulations to constrain past aerosol radiative forcing changes (e.g. Smith et al., 2021). Since the simulations do not include different spatial and temporal uncertainties in emissions, they are not including the real uncertainty in the aerosol forcing. Other studies use CMIP6 or similar simulations to attribute the change in temperatures or precipitation to different forcings (e.g. Biasutti and Giannini,

2006; Undorf et al., 2018; Hegerl et al., 2019), and these attempts to attribute changes could be based on a poor representation of the real uncertainty in the preindustrial of the aerosol forcing. Aerosol radiative forcing uncertainty cannot be constrained easily by temperature time series, since other uncertainties can be difficult to pull apart from aerosol uncertainties (Kiehl, 2007; Lee et al., 2016).Attributing climate at a regional scale is the next frontier of detection and attribution, but this cannot be done without accurate aerosol histories (Lehner and Coats,

490    2021).





In summary, we argue that it is critical that the full uncertainty deduced in the aerosol literature, including due to changes in natural aerosols, be passed to physical climate scientists so that they can accurately account for these in constraints on climate sensitivity and in projections of future climate changes.


## 8        Conclusions: Pathway to improve historical and paleoclimate characterization of uncertainties

How can we address the systematic underestimate in the uncertainty of changes in aerosol radiative effects between different time periods? Here we propose some steps towards first characterizing the true uncertainties, including

emission uncertainties, and then using observations to constrain these aerosol pathways to constrain the radiative forcing and uncertainties.

a. Characterize historical uncertainties in aerosol and aerosol precursor emissions, using available knowledge of emissions, and how they might have changed. These estimates should include some versions which are

consistent with available paleodata (Kok et al., 2023; Hamilton et al., 2018). The uncertainties from emissions should be combined with uncertainties in aerosol processes to create a more robust uncertainty bound for different time periods.

b. Characterize paleoclimate emissions of aerosols and the resulting radiative forcing at important past climates, such as last glacial maximum and last interglacial. These estimates should be based as much as possible on

observations, and possible ranges.

c. We propose a new intercomparison project (AEROHISTMIP) which would include multiple emission pathways in the historical model simulations conducted for CMIP exercises. Some models should not only conduct ensemble members of one aerosol emission scenario, but use multiple aerosol emission scenarios to better understand the resulting uncertainty in aerosol radiative forcing and climate response to likely emissions in

different time periods. The evolution of several related past model intercomparison projects under one umbrella (e.g. Cloud Aerosol and Terrain Interactions: CACTI) provides the ideal opportunity now to include such simulations.

d. Constrain preindustrial to present day aerosol radiative effects. From (c), combined with observations (a), the most likely past emissions scenarios can be identified, and we can make the first steps towards constraining

uncertainty, similar to the efforts underway to characterize which of the climate models are most reliable (e.g. IPCC, 2021).



e. Obtain more paleoclimate proxies for aerosol concentrations. Here we have focused mostly on wildfires and dust, since there is enough paleoclimate data to show that CMIP6 does not represent historical changes in these aerosols well, but indeed it is not possible currently to validate the changes in emissions for other natural
(e.g., sea spray) and anthropogenic (e.g., sulfate) aerosols as well. We need the development of more proxies for historical and paleoclimate changes in aerosols to increase confidence in our estimates of aerosol radiative forcing.

f. Continue to improve aerosol measurement databases, including more in situ observations of the aerosol composition in more locations (Snider et al., 2016), as well as continued use of satellite observations to
constrain the magnitude of aerosol radiative forcing (e.g. Smith et al., 2021). We encourage more observations for variables directly related to the radiative forcing (e.g., aerosol optical depth) and those that could help narrow uncertainties in crucial parameters that describe related physiochemical processes. Some of these data can also be used to assess the model performance, narrowing the model spread, and validate satellite retrievals. As aerosol number concentration, the determinant of the change in cloud properties to emission
changes, cannot be retrieved from paleo proxies (d) there needs to be a simultaneous effort in understanding natural aerosol processes and impacts on clouds under pristine "PI-like" present day conditions (Hamilton et al., 2014; McCoy et al., 2020)

g. Characterize current model direct aerosol radiative effects and aerosol cloud interactions using new tools. Currently meteorological models are tightly connected to the aerosol models they host, making it difficult to
independently evaluate the structural differences between aerosol models. Recent efforts to develop generalized chemical and aerosol interfaces would allow more effective evaluation of chemical and aerosol schemes separate from their host models (Hodzic et al., n.d.). A generalized framework could also allow artificial intelligence methods to be integrated into multiple models and model-data comparisons or assimilations to be used across models.


In summary: while there has been substantial progress in aerosol-climate science over the past 20 years, aerosols remain one of the most important uncertainties in climate change science, and are likely to continue to be important to study for at least the next 20 years.

**Data Availability**

N/A



## Author contribution

NMM conceived of the idea and created the first draft, with substantial improvements from LL, SA and JFK.

## Competing interests

None


## Acknowledgements

NMM and LL would like to acknowledge support from DOE DE-SC0021302. JFK acknowledges support from NSF grants 1856389 and 2151093. We acknowledge helpful discussion with Flavio Lehner and Steve Sherwood.

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



**Figure captions**

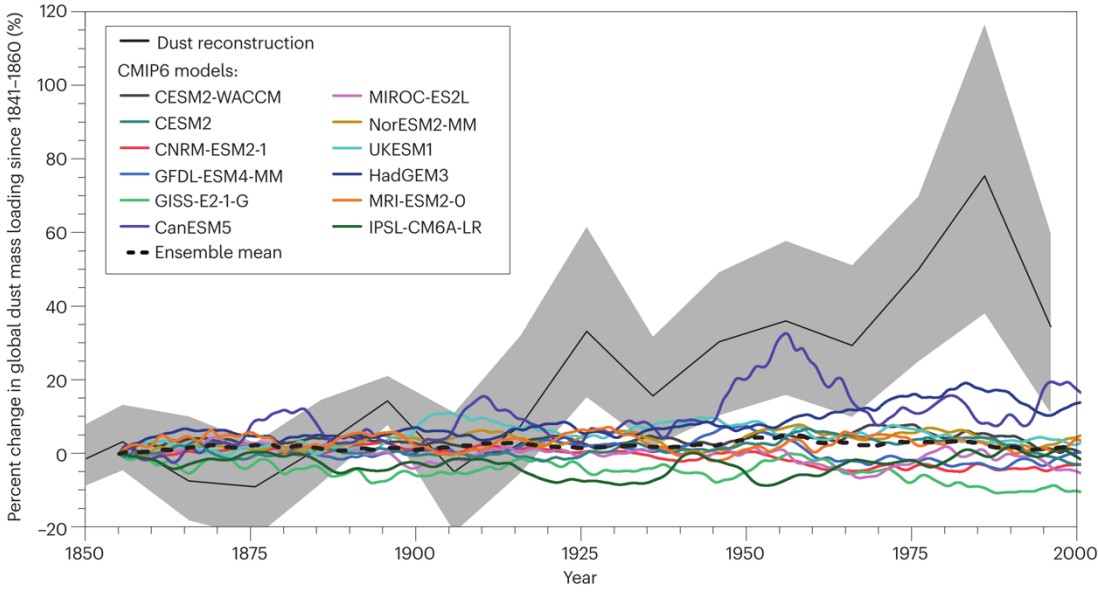

Figure 1. Annual global mean dust aerosol loading from the observationally based reconstruction (Kok et al., 2023) in black, with grey error bars (representing the 90% confidence interval), compared against 10-year running means from historical runs of Climate Model Intercomparison Project (CMIP6) ensemble members. Reproduced with permission from (Kok et al., 2023).



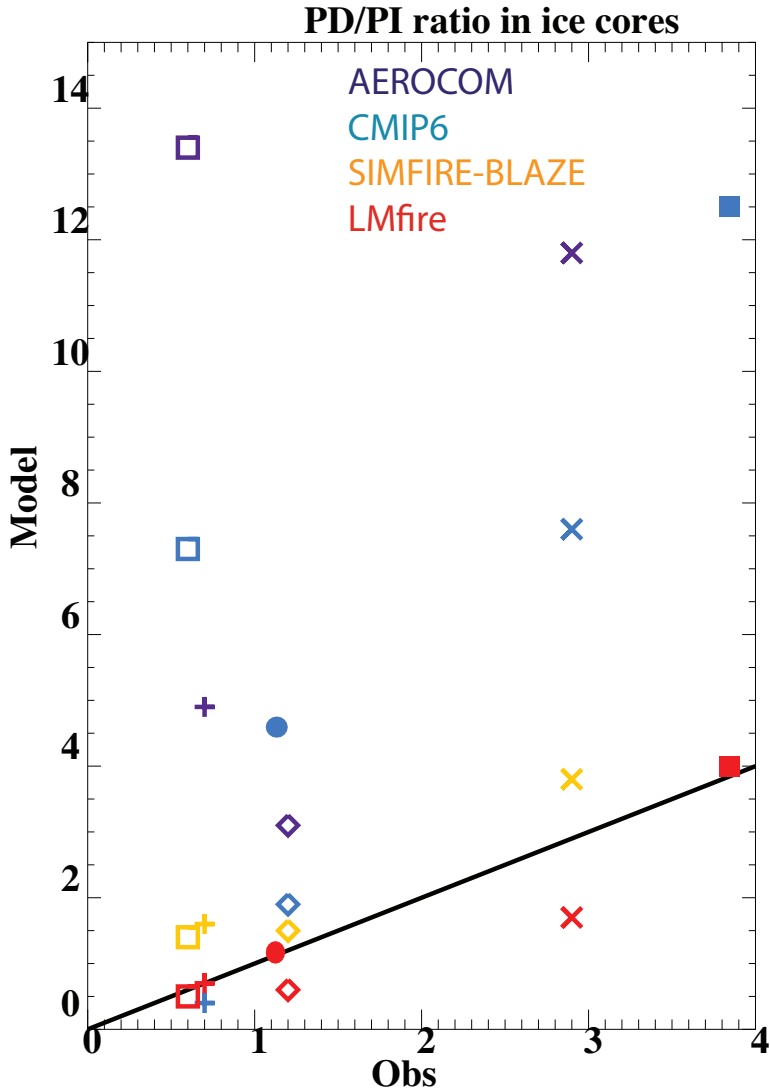

Figure 2: Observed and modeled present day to preindustrial ratios (PD/PI) for black carbon in ice cores using different open fire emissions. Ice cores sites are from Greenland (open square and plus signs), Wyoming (diamond) and France (x's) for 4 different model simulations: AEROCOM (purple), CMIP6 (blue), SIMFIRE-BLAZE (yellow) and LMfire (red) are taken from (Hamilton et al., 2018). Ice core sites from Bolivia (solid circle) and Antarctica (solid square) using CMIP6 (blue) and LMfire (red) are taken from (Liu et al., 2021). The solid black line shows the 1:1 line.



## Full uncertainty of aerosol changes requires multiple emission pathways and comparisons to observations

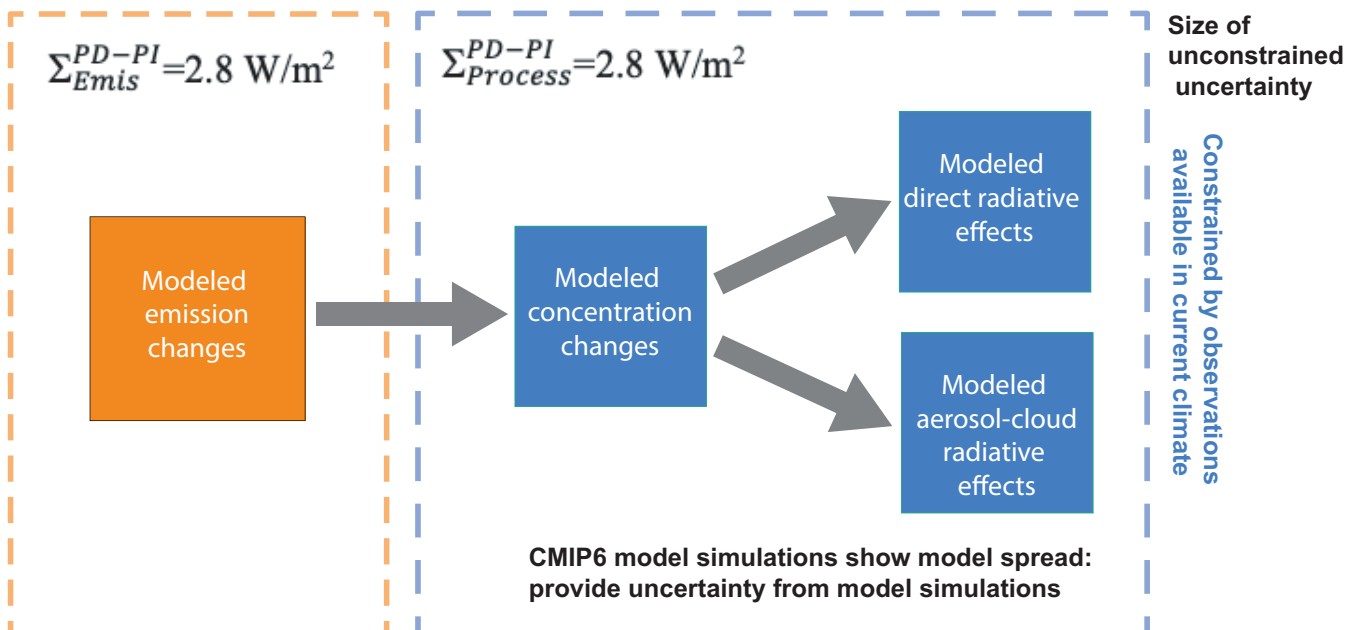

Figure 3: Schematic of the sources of uncertainties in aerosol radiative effects, from emission models to modeled concentration changes to modeled direct and aerosol-cloud radiative effects. The CMIP6 unconstrained uncertainties using a single emission scenario have a 90% confidence interval range of 2.8W/m$^2$ (Bellouin et al., 2020; Sherwood et al., 2020). For the uncertainty using different emission scenarios for the past climate, the 90% confidence interval ranges from wildfires of 2.8 W/m$^2$ (Hamilton et al., 2018; Wan et al., 2021) is added to uncertainties from dust of 0.4 W/m2 (Kok et al., 2023) and added to an estimate of industrial emission uncertainties (assuming 10% error) of 0.2 W/m$^2$. We square these errors and take the square root to obtain 2.8 W/m$^2$ uncertainty in emissions.



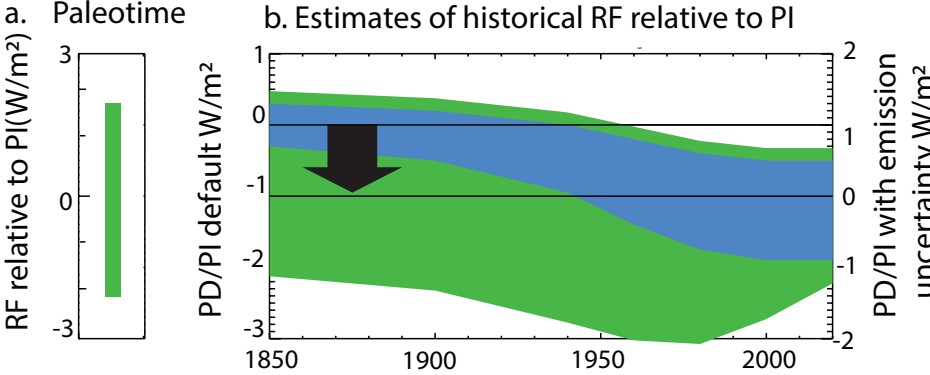


Figure 4: Estimates of the unconstrained radiative forcing of aerosols and uncertainties for (a) different paleoclimate time periods relative to preindustrial (1850) and (b) radiative forcings and uncertainties relative to 1850 based on CMIP6 model spread (schematic based on (Smith et al., 2021) in blue) and including the emission uncertainties (90% confidence intervals) from wildfires, dust and anthropogenic aerosols as described in Fig. 3 (green), using the time series for wildfires from (Marlon

et al., 2008). The left vertical axis represents the present day minus preindustrial radiative forcing following (Smith et al., 2021; Sherwood et al., 2020), and the right axis adds in the emission uncertainties for the preindustrial (from Fig. 3; Hamilton et al., 2018), shifting the preindustrial baseline (black arrow). Notice that the size of the black arrow and shift in the preindustrial state is not known, and this is a schematic to illustrate how the uncertainties in emissions in the preindustrial impact understanding of the radiative forcing.



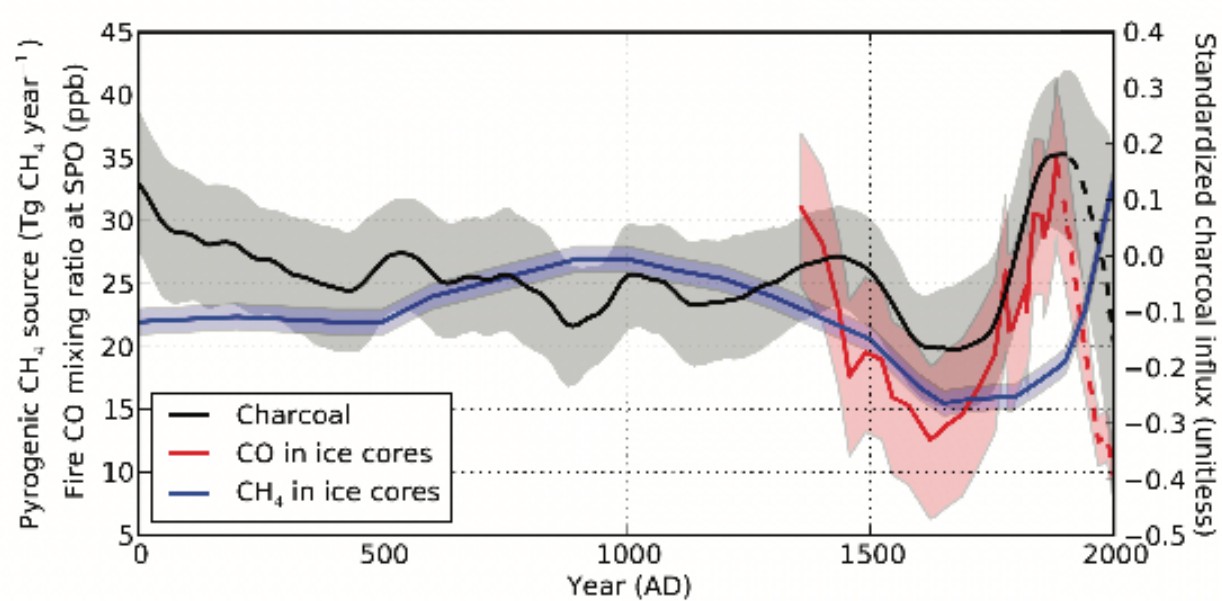


Figure. 5. Variability of biomass burning rates over the last centuries based on a worldwide compilation of charcoal records (Marlon et al., 2008), CO mixing ratios from fires using CO concentration measurements at the South Pole (SPO), its isotopic signature, and a mass balance model (Wang et al., 2010) and a similar approach but based on $CH_4$ (Ferretti et al., 2005). The CO ice core data ended in 1897 but were extended (dashed line) by Wang et al. (2010) to present-day using firn samples (1968

and 1986) as well as modelling (year 2000). Shaded areas indicate reported uncertainty. Note that the datasets have different footprints and that absolute values cannot be compared directly.  Reproduced with permission from (van der Werf et al., 2013) under CCC3.0.




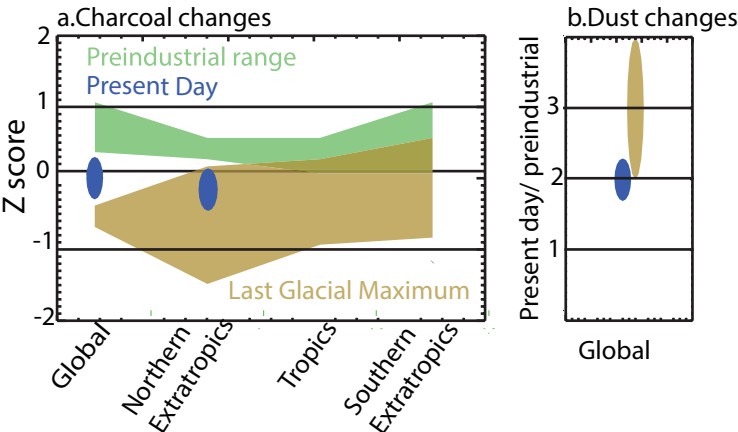

Figure 6: Relative size of paleoclimate and historical changes in aerosols. A) Based on z scores from charcoal records, the variability across preindustrial time periods (green), present day (blue) and last glacial maximum is shown in global, northern extratropics, tropics and southern extratropics based on data from (Marlon et al., 2008, 2016). Charcoal reconstructions use z-scores, which normalize around the mean value at a site, divided by the variability, and thus a -2 z-score for LGM suggests significantly lower charcoal amounts. B) global dust changes ratio of deposition between present day and preindustrial (blue oval; (Mahowald et al., 2010; Kok et al., 2023), and for the last glacial maximum relative to preindustrial (gold oval) (Mahowald et al., 1999; Albani et al., 2014, 2018; Lambert et al., 2015).



**Table 1.** Estimates of aerosol radiative forcing uncertainty, symbols, and citations. These represent the 90% confidence intervals.

| Aerosol radiative forcing uncertainty term | Symbol | Estimate (range of 90% confidence interval) | Source |
|---|---|---|---|
| Unconstrained process uncertainty: Present day to preindustrial unconstrained model spread with fixed emissions from CMIP6 | $\Sigma_{Process}^{PD-PI}$ | 2.8 W/m$^2$ | (Bellouin et al., 2020; Sherwood et al., 2020) |
| Unconstrained emission uncertainty: Present day to preindustrial uncertainty in emission changes unconstrained | $\Sigma_{Emis}^{PD-PI}$ | 2.8 W/m$^2$ | Equation 1 and. (Hamilton et al., 2018; Wan et al., 2021; Kok et al., 2023) |
| Total unconstrained uncertainty: Present day to preindustrial | $\Sigma_{Total}^{PD-PI}$ | 4.0 W/m$^2$ | Equation 2 |
| Constrained process uncertainty: Present day to preindustrial constrained with observations using CMIP6 emissions | $\Sigma_{Const\_Process}^{PD-PI}$ | 1.6 W/m$^2$ | (Bellouin et al., 2020) |
| Unconstrained emission uncertainty: preindustrial time period | $\Sigma_{Emis}^{PI}$ | 2.8 W/m$^2$ | PI uncertainties in emission drive uncertainties in PD-PI |



| Unconstrained process uncertainty: Paleotime T to preindustrial | $\Sigma_{Process}^{T-PI}$ | 2.8 W/m$^2$ | Assume same as PD to PI |
|---|---|---|---|
| Unconstrained emission uncertainty: Paleotime T to preindustrial uncertainty | $\Sigma_{Emis}^{T-PI}$ | 4.0 W/m$^2$ | Equation 4 |
| Unconstrained total uncertainty: Paleotime T to preindustrial uncertainty | $\Sigma_{Total}^{T-PI}$ | 4.8 W/m$^2$ | Equation 3, using $\Sigma_{Emis}^{T-PI}$ and $\Sigma_{Process}^{T-PI}$ |

985