# Peer review of "Opinion: The importance of historical and paleoclimate aerosol radiative effects"

_EGUsphere, 2023_

## Author Comment (AC1)

We would like to thank the reviewers for their constructive criticism to improve the manuscript. Original comments from the reviewers are in black and our responses are in red.

Reviewer 1:

This is a nice summary of what is known about natural sources of aerosols and their uncertainties. I recommend publication. I unfortunately was unable to get a copy of the Kok et al. 2023 preprint, so I could not check on many of the facts noted in this paper which were based on that paper. However, assuming it is well written, I have no problem with publishing this.

Thank you for your helpful comments. It is unfortunate that the Kok et al., 2023 review article was hard to get, but it is now available at: https://jasperfkok.files.wordpress.com/2023/07/kok_et_al_2023_nature_reviews_dust_impacts_on_climate.pdf

I note the follow which should be clarified:

Line 47: Van Marle et al 2017 is not in the reference list.

Thank you for this point: we add this paper to the reference list.

Line 72-74: point 1 mentions natural aerosol feedbacks, and this is repeated in point 2

We agree: we remove it from point 1.

Line 81: you mention "among others", which made me think about DMS (sulfate), NOx lightning (nitrate). Might be better to list a few of these, even though you don't really discuss them in the paper.

We add in sulfate and NOx as examples.

Line 133: process rather than processes? Or restate

We agree: process is correct.

Line 153: change "that is" to "that it is"

Changed.

Line 210: you mention the timecale in (ii) and again in (iii). If there is a different point you are trying to make, please explain.

We agree that it makes sense to mention timescale only in (iii).

Line 221-222: why do you conclude that it is best to treat the historical dust change as a forcing when you don't know whether it is due to a forcing or a feedback. This should be clearly argued as opposed to just saying it should be treated as a forcing when you don't know.

We try to rephrase this to be more clear by changing these lines to:

New lines 235-239:

Considering these important issues discussed above, we argue that model simulations should include historical dust changes as an external forcing for two reasons: 1) because it is unlikely that the anthropogenic forcing and the climate feedback components of the historical dust loading change can be reliably separated in the near future and 2) because climate models currently cannot reproduce the historical dust changes (Kok et al., 2023; Mahowald et al., 2010).

Line 931: should "ranges" be "range"?

Corrected.

Line 248: Reword/Expand on the explanation that much of anthropogenic radiative forcing is from fires and how this explains the large emission uncertainty from fires.

We try to clarify (new lines 263-267)

"If we sum the sources of uncertainty currently available in the literature ($\Sigma_{Fires}^{PD-PI}$ =2.8 W/m$^2$ from fires, $\Sigma_{Dust}^{PD-PI}$ =0.4 W/m$^2$ from dust (Hamilton et al., 2018; Wan et al., 2021; Kok et al., 2023) and here we assume a 10% error for industrial emissions for $\Sigma_{Industry}^{PD-PI}$=0.2 W/m$^2$ using Eq. (1), we obtain 2.8 W/m2, clearly dominated by fires (see Table 1 for terms). This is consistent also with attribution literature, which shows that much of the anthropogenic radiative forcing is from the biomass burning source (Figure 1a in Unger et al., 2010).

"

Line 311-313: I think you mean to say that estimates of anthropogenic aerosol radiative forcing today would be smaller. If I'm not correct, please explain more thoroughly.

We try to make this more clear:

New lines 331-334: "Thus, if we take the case of high wildfires in the preindustrial (-2 W/m$^2$ in Fig. 4), this could imply that estimates of aerosol radiative forcing from purely anthropogenic sources today which are large (-2W/m$^2$) would be more likely; the 1850 to 2000 aerosol radiative forcing would be the same as assumed now, but the 1850 aerosol radiative forcing would already be large."

Line 961,962: the gold oval makes sense if you read the explanation for B) (i.e. the last glacial maximum had higher dust), but not if you look at the y-axis (the gold oval is not the present day/preindustrial or present day/last glacial maximum)

Agreed: we correct.

Line 353: change describe to described; also, why wouldn't increased knowledge reduce these uncertainties? Please clarify.

We try to be more clear:

New lines 375-377: "The uncertainties in emissions in PI are driving the uncertainties in PD-PI describe above and are about $2.8W/m^2$ ($\Sigma_{Emis}^{PI}$): since these uncertainties are due to variability not a lack of information, increased knowledge is unlikely to reduce these uncertainties (only if the new studies show less variability). "

Line 401-402: are you missing a word? Or should "one of the largest aerosols" be "one of the largest uncertainties"

Agreed: "aerosols" should be "uncertainties"

Line 516: CACTI stands for Composition, Air quality, Climate inTeractions Initiative

Thank you, we correct.

Reviewer 2:

The authors highlight the uncertainties in evolution of aerosol radiative forcing, particularly due to aerosols from natural systems (using dust and wildfire as example) and review the current status of paleo observations to constrain past modeled aerosols and their radiative effects. They argue for dust and wildfire emissions to be considered as external forcing driving the climate system rather than as feedbacks. The primary premise of this paper to recognize and quantify the cascade of uncertainties in aerosols radiative effects beginning from emissions to radiative effects. I think this will be a very useful review for the community. I recommend publication after consideration of my comments below.

Thank you for your constructive suggestions.

L26: This may work in theory but we know that there is variation across models in the simulation of aerosols even with the same emissions dataset. How would good model-obs comparison of one model with one emissions dataset be reconciled with another model using another emissions dataset?

We agree with the reviewer. We propose that several Earth system models be used with several emission datasets. We don't think this clarification should be in the abstract, but add it into the conclusion more clearly.

New lines 538-541:"Several Earth system models should not only conduct ensemble members of one aerosol emission scenario, but also use multiple aerosol emission scenarios to better understand the uncertainty in aerosol radiative forcing and climate response due to the uncertainties in aerosol emissions in different time periods. "

L33-35: It would be helpful to cite the specific IPCC 2021 chapter(s) being referred to here and throughout the paper.

We actually are citing the summary for policymakers in the reference list, but we agree that the underlying chapters could be more useful to be cited directly, and have added the two IPCC state of the climate and short-lived chapter citations as suggested below.

L43-45: Following on from the previous sentence, it may be more logical to discuss uncertainties in these natural emissions and then relate to uncertainties in radiative forcing.

Good point. We rewrite the sentence as follows:

New lines 56-59:"On the other hand, the uncertainty in aerosol emissions that are usually considered natural, such as from dust storms or wildfires, are likely larger, and contribute to larger uncertainties in aerosol radiative forcing (Carslaw et al., 2010; Mahowald et al., 2011a; Regayre et al., 2018). "

L45-47 – This sentence is focused on past climates but is referencing Gidden et al which deals with future projections of emissions.

We remove reference to Gidden et al. and replace with Hoesly et al.

L50-59 – Note that most ESMs in CMIP6 include interactive representation of many natural aerosol emissions or their precursors (e.g., dust, DMS, sea-salt, BVOCs…). This implies that constraining models' past emissions of natural aerosols would require constraining the parameterizations to the limited and uncertain paleo-observations. And it is possible that when constrained to paleo obs, these parametrizations may not represent the modern day emissions (as evaluated against current observations). It would be helpful if authors could shed light on this catch-22 situation.

This is a good point. We don't think the introduction is the place to go into this, but added the following text in the forcer or feedback section:
New lines 331-335: "Since many models include prognostic schemes for dust, we propose that models add a temporally varying emissions factor obtained from constraints on the historical evolution of atmospheric dust deposition (e.g., Mahowald et al., 2010). This would enable models to both reproduce the historical change in dust, yet to also predict future changes in dust forced by climate and land use changes. A similar approach could be used for other natural aerosol emissions, such as from wildfires."

And in the conclusion section when we talk about forcing the models to match observations:

New lines 662-665: "Note that many 'natural' aerosols are prognostic in the models (e.g. dust), and therefore to in order to match available scenarios, the prognostic aerosol schemes may require to be corrected using a temporally varying emission factor to simulate the correct temporal trends (e.g. Mahowald et al., 2010)."

L66-67 – My understanding is that attribution of forcing (or specifically climate change) is needed to inform climate change mitigation policies. It is therefore important to attribute the

radiative perturbations in aerosols to either natural processes or human activities. However, quantifying the extent of human modification of dust, wildfire or any naturally occurring process emissions is difficult and is largely uncertain (the authors note this difficulty on L180-L183). This then translates into large uncertainties in the attribution of forcing for natural system emissions perturbed by human activities. In principle, I agree with the authors' argument that forcing due to perturbations in any natural system emissions modified by human activities should be quantified but I am not convinced that we have reached a point in our state of knowledge to be able to do this without large uncertainties.

We agree completely with the reviewer, and this a main theme of this article. We add the following to the conclusions:

New line 679-681 "Any inability of existing models to simulate observations, as well as other remaining uncertainties, should be carefully assessed: it may not be possible for the models to simulate the observed changes."

L74 – note that the CMIP6 emissions did not include dust emissions

This is an important point that we think is covered by our response to the reviewer's comment about how to get prognostic aerosols to be 'forced'.

L99-100 – Note that there are proxies for other aerosols such as black carbon and sulphate assessed by Gulev et al. (2021) (section 2.2.6).

We add a reference to this section of the Gulev et al., 2021 chapter in lines 190, where we discuss the uncertainties also in fossil fuel sources and how few observations there are in Figure 2.9a and 2.9b to compare against available observations.

 IPCC AR6 WGI Chapter 6 should be cited as Szopa et al (2021) here and throughout the paper.

We add a citation of these chapters.

L122-123 – What drove this large increase in dust – land use changes, climate change or both?

No one knows for sure: We discuss this in Section 3, so in order to not be repetitive we don't add it here.

L125-128: All models or a subset? There were a number of models that prescribed aerosol properties to capture the influence of aerosols on climate. These models presumably did not simulate dust.

Good point. We correct to:

New lines 160-164: "Unfortunately, the model simulations from the last Climate Model Intercomparison Project (CMIP6) that include prognostic dust do not match these changes, either because of the lack of

correct inclusion of the impact of land use onto dust, or incorrect simulation of the feedbacks of natural and anthropogenic climate changes onto dust (Fig. 1). Indeed, the CMIP6 models show relatively constant dust amounts in contrast to the large change that is observed (Kok et al., 2023)."

L148-153 – sentence is too long, revise. Define AEROCOM and provide a reference for the emissions dataset

We correct to:

New lines 185-190: Indeed, those fire models which include a more realistic representation of how fires and human population density are related simulate a much higher amount of fires during the pre-industrial (c. 1850) than the CMIP6 (or AEROCOM) emission dataset suggests (Hamilton et al., 2018; Hoesly et la., 2018; Dentener et al., 2006). Including larger past fire emissions in aerosol models also improves the match of simulated data to the available ice core data of the deposition ratio between present day to preindustrial of black carbon (yellow and red symbols) than using the default CMIP6 datasets (blue symbols) (Hamilton et al., 2018; Liu et al., 2021) (Fig. 2).  This suggests that it is probable that there were more open fire emissions during the preindustrial  than accounted for in the emission inventories used for CMIP6 simulations (Hoesly et al., 2018; van Marle et al., 2017).

L155 – is Gidden et al the correct reference here?

Thanks for pointing this out: we switch to Hoesley et al., 2018.

L159-160 – This reasoning is not clear to me – how could higher fire amounts during 1850 be due to less land use change? Wouldn't land-use require clearing of land and therefore more fires?

Yes, the text is unclear. We rewrite to:
New lines 221-224: "One study suggests that higher fire amounts in 1850 relative to today could be due to more land use change today than in 1850.  In other words,  today there is less  natural area available for fires than 1850 (van der Werf et al., 2013) which is consistent with the decreased fire burn area observed over the satellite era (Andela et al., 2017)."

L172-174 – Please specify the IPCC 2021 chapter that assessed the radiative effect of fire aerosols to be -2 Wm-2. Are there any uncertainties associated with this estimate? Ditto for IPCC 2019 and -1Wm-2.

The manuscript cited the SPM, but we add citations of the chapters as suggested by the reviewer. We clarify to:

Lines 233-237: In addition, wildfires and open fires represent some of the most important aerosols for direct and aerosol-cloud radiative effects, with a total radiative effect in the current climate of -2 $W/m^2$ (IPCC, 2021; Gulev et al., 2021; Szopa et al., 2021; Penner et al., 1992).  Changes in wildfires and open fires represent about -1 $W/m^2$ or 50% of the anthropogenic aerosol radiative forcing since 1850, and all of these estimates have very large uncertainties (Unger et al., 2010; IPCC, 2019, 2021; Gulev et al., 2021; Szopa et al., 2021).

L189-192 – Although the radiative forcing due to changes in dust emissions is not explicitly accounted for, the IPCC assesses the influence of human activities on emissions and the large

associated uncertainties (section 6.2.2.4) – "In summary, there is high confidence that atmospheric dust source and loading are sensitive to changes in climate and land use, however, there is low confidence in quantitative estimates of dust emission response to climate change."

We add this point:

New lines 285-288: "For instance, in the Sixth Assessment Report, the radiative perturbation due to the ~50% increase in dust over the historical record (Hooper and Marx, 2018; Mahowald et al., 2010; Kok et al., 2023) is not explicitly accounted for as a radiative forcing of the climate system, although the dust-climate feedback was quantified (Forster et al., 2021; Naik et al., 2021) and the report does highlight that there is substantial uncertainty in this feedback (Szopa et al., 2021)."

L223-225 – It would be also helpful to recommend a specific dust emissions dataset that the modelers could use to prescribe dust emissions. This could inform the CMIP7 process.

Good point.  We point out in the conclusion, section a, the reference to Kok et al., for the dust, although as discussed before, this should be a combination of forcing and prognostic aerosol.

New lines 658-662: "Characterize historical uncertainties in aerosol and aerosol precursor emissions, using available knowledge of emissions, and how they might have changed.  These estimates should include some versions which are consistent with available paleodata (e.g. for dust Kok et al., 2023; and for wildfires Hamilton et al., 2018 and Liu et al., 2021).  "

L245-249 – It would be helpful to place the relevant citation next to the uncertainty estimate from the literature so that the source of these numbers is clear. What is the source of the 10% error for industrial emissions? Note that the Unger et al reference is now 13 years old…emission estimates have changed, models have changed. Any updates to that study?

We just assume 10% error: must be at least that. We move the parenthesis to make it more clear that there is no citation for that error estimate, but is just assumed here.  We do not know of an update to Unger et al.

New lines 373-377: "If we sum the sources of uncertainty currently available in the literature ($\Sigma_{Fires}^{PD-PI}$ =2.8 W/m$^2$ from fires, $\Sigma_{Dust}^{PD-PI}$ =0.4 W/m$^2$ from dust (Hamilton et al., 2018; Wan et al., 2021; Kok et al., 2023)  and here we assume a 10% error for industrial emissions for $\Sigma_{Industry}^{PD-PI}$=0.2 W/m$^2$ using Eq. (1), we obtain 2.8 W/m2, clearly dominated by fires (see Table 1 for terms).  This is consistent also with attribution literature, which shows that much of the anthropogenic radiative forcing is from the biomass burning source (Figure 1a in Unger et al., 2010)."

254 – "…emission scenario for the historical period…" it would be better to replace scenario with another word to avoid confusion with future scenarios.

We disagree: we do not know what will happen in the future and we do not know what happened in the past. We need a term to indicate that, which scenario does. We add a clarification:

New line 394-396: "Note that here we use the same term 'scenario' for what happened in the past as what we use for choices in the future, to emphasize that we do not know these past emissions. "

L281 – Gidden et al should be replaced with van Marle et al (2017).

We agree: we correct this.

L402 – "aerosol uncertainties remain one of the largest aerosols in those times…" largest aerosols?

Corrected: should be uncertainties.

L511 – The success of the proposed AEROHISTMIP will depend on the availability of multiple emission realizations for the historical period. It would be helpful to provide some indication of how these datasets will be put together and who would be responsible for making the files available to the CMIP effort. Without this information, I don't see this recommendation leading to a tangible action. Additionally, if specific simulations are being suggested, I would recommend adding them to the AerChemMIP2 (https://airtable.com/shrtJ4jc08OEk7Vcq/tblAfxwzZTy4soluj) effort rather than a new intercomparison.

It will actually take the community some time to come up with new emission datasets that span the whole observational uncertainty, although we have proposed here some ideas. It is beyond the scope of this paper to solve this problem. We add this point to the text.

New lines 670-671: "Note that developing these new emission pathways is likely to be beyond one or two group's capabilities and thus may require a workshop or other community activity."

L518-521 – This goes back to my earlier point, models vary in their representation of aerosol processes in part driven by lack of full understanding of the various processes that determine the evolution of their atmospheric burden and radiative effects. With such gaps in process understanding can we really characterize the reliability of models for aerosols? I think some thought needs to be given to this recommendation.

We clarify: if we are unable to constrain the past emissions well enough, we need to make sure that the uncertainties are accurately assessed and passed to the physical climate scientists.

New lines 680-681:"Any inability of existing models to simulate observations, as well as other remaining uncertainties should be carefully assessed."

L542 – It is not clear if Hodzic et al is in preparation or submitted?

We clarify: it is submitted.

Figure 3 – It should be noted somewhere on the figure that $\sum$ represents uncertainties to avoid confusion.

Excellent point: corrected.

Finally, the paper needs a thorough proof-read and editing to improve the quality of text.

Thank you for your helpful comments: we have carefully reviewed the text for errors.

References:

[revised manuscript text omitted]